# Facile Synthesis of CoOOH Nanorings over Reduced Graphene Oxide and Their Application in the Reduction of p-Nitrophenol

**DOI:** 10.3390/ma15248862

**Published:** 2022-12-12

**Authors:** Huihui Chen, Mei Yang, Jun Yue, Guangwen Chen

**Affiliations:** 1School of Chemistry and Chemical Engineering, Henan University of Technology, Zhengzhou 450001, China; 2Dalian Institute of Chemical Physics, Chinese Academy of Sciences, Dalian 116023, China; 3Department of Chemical Engineering, Engineering and Technology Institute Groningen, University of Groningen, 9747 AG Groningen, The Netherlands

**Keywords:** cobalt oxide, reduced graphene oxide, nanorings, catalysis, p-nitrophenol, reduction

## Abstract

A facile and one-step route has been employed for the synthesis of highly uniform CoOOH nanorings assembled on the surface of reduced graphene oxide (CoOOH/rGO nanocomposite). The physicochemical properties of the obtained CoOOH/rGO nanocomposite were characterized using X-ray diffraction pattern (XRD), scanning electron microscopy (SEM), transmission electron microscopy (TEM), N_2_ physical adsorption (BET) and X-ray photoelectron spectroscopy (XPS). The TEM and SEM results confirmed that CoOOH nanorings (edge length ∼ 95 nm) were uniformly decorated on reduced graphene oxide nanosheets using the simple precipitation–oxidation–reduction method. When used as a catalyst for the reduction of p-nitrophenol to p-aminophenol in the presence of excess NaBH_4_, the resulting CoOOH/rGO nanocomposite exhibited good activity and stability. When the initial concentration of p-nitrophenol was 1.25 × 10^−4^ mol·L^−1^, p-nitrophenol could be fully reduced within 3.25 min at room temperature. The apparent rate constant was estimated to be 1.77 min^−1^, which is higher than that of pure CoOOH nanorings. Moreover, p-nitrophenol could still be completely reduced within 6 min in the fifth successive cycle. The superior catalytic performance of the nanocomposite is attributed to the synergistic effect between the highly dispersed CoOOH nanorings and the unique surface properties of the reduced graphene oxide nanosheets, which greatly increased the concentration of p-nitrophenol near CoOOH nanorings on reduced graphene oxide surface and improved the local electron density at the interface.

## 1. Introduction

With the rapid development of the chemical industry, water pollution caused by mass production of various chemical products is increasingly aggravated. Among them, organic pollution is a non-negligible factor, which seriously threatens the ecological environment and human health. p-Nitrophenol (p-NP), one of the most hazardous organic pollutants in industrial and agricultural wastewater, is mutagenic and resistant to biodegradation [1]. Its residue and accumulation in the environment may damage the liver, kidney, and nervous system, and even cause blood disorder or cancer in humans [2,3,4]. It has been classified as a highly hazardous contaminant and its maximum concentration should be below 10 ng/L in natural water, according to the U.S. Environmental Protection Agency. Therefore, the elimination or degradation of p-NP using a cost-effective and highly efficient method is highly significant for mitigating its adverse impact on the environment. Up to now, the catalytic reduction of p-NP to p-aminophenol (p-AP) using NaBH_4_ as a reducing agent is considered to be a promising and efficient strategy, because the reduction process produces only one product p-AP, which is less toxic and has a higher allowable concentration in the environment [5]. Furthermore, p-AP is a highly useful fine chemical and pharmaceutical intermediate, e.g., in the production of antipyretic analgesics, paracetamol, vitamin B_1_ and dyes [6].

Due to the high stability of nitrophenol compounds, appropriate catalysts are required to lower the kinetic barrier and accelerate the reaction rate for the reduction of p-NP to p-AP [7]. Various noble metal nanoparticles (like, Au, Ag, Pd, Pt, etc.) have been widely studied in recent years due to their excellent catalytic activity in the reduction of p-NP [8,9,10,11]. However, noble metal nanoparticles tend to seriously aggregate due to their small particle size during the reaction, and their high price due to limited reserves restrains their large-scale applications [3,12]. With the rapid development of nanotechnology, transition metal oxide nanomaterials have received extensive attention as catalysts, providing the possibility to substitute some noble metal catalysts. It has been reported that several transition metal oxide-based heterogeneous catalysts, such as Co_3_O_4_, CuO, Fe_2_O_3_ and NiO [13], exhibit catalytic activity in the reduction of p-NP to p-AP. For example, Che et al. [14] found that CuO nanostructures showed good catalytic activity for p-NP reduction and their catalytic activity in NaBH_4_ aqueous solution was dependent on its morphologies. Mogudi et al. [15] reported that mesoporous cobalt oxide could be used as a cost-effective catalyst for p-NP reduction. Moreover, Elfiad et al. [16] demonstrated the good catalytic activity and stability of natural α-Fe_2_O_3_ for the reduction of p-NP. 

In particular, metallic cobalt and cobalt oxide catalysts have been considered as prospective candidates for the reduction of p-NP due to their high activity and low cost. As an example, magnetic Co/EAtp@C (carbon-coated etched attapulgite) nanocomposite used as a catalyst could lead to almost complete reduction of p-NP to p-AP within 8 min at room temperature [17]. Co-carbon composite containing N species, synthesized by one-step calcination of cobalt-based metal organic frameworks, could reduce p-NP to p-AP by nearly 100% within 115 s at room temperature [18]. Co_3_O_4_ nanoparticles are also frequently studied as catalysts for the reduction of p-NP to p-AP by NaBH_4_. Co_3_O_4_/BiFeO_3_ nanocomposite has been confirmed to be an efficient and recyclable catalyst for the reduction of nitrophenol isomers [19]. Our previous work has demonstrated that the catalytic activity of Co_3_O_4_ for the reduction of p-NP to p-AP was remarkably improved by the introduction of oxygen vacancies, and its apparent rate constant reached 1.49 min^−1^ at room temperature [20]. As a common cobalt oxide, CoOOH has a wide range of applications in sensors [21], supercapacitors [22], lithium-ion batteries [23], and catalysis. For example, it can be used as a catalyst for water oxidation [24]. However, there is no report on CoOOH as a catalyst to catalyze the reduction of p-NP to p-AP.

To further enhance the catalytic performance of cobalt-based catalysts, many strategies have been investigated, such as morphology modulation [25], element doping [18], surface defects fabrication [20], pores introduction [26], and support-induced interactions [27,28]. Among these modification methods, the immobilization of cobalt-based catalyst on an efficient support has received extensive attention because of its ability to prevent nanoparticle aggregation and the possible synergistic effect, which is beneficial to improve their catalytic activity [29,30]. For instance, halloysite nanotubes supported Co_3_O_4_ nanoparticles (Co_3_O_4_/HNTs) showed good catalytic performance because of the specific structural properties of Co_3_O_4_ endowed by halloysite nanotubes and the cooperative effects between them [30]. Owing to the synergistic effects between the support and metal-containing phase, commercial sponge-derived Co_3_O_4_@C catalysts exhibited excellent catalytic properties in styrene oxidation, rhodamine B decolourization and p-NP reduction [31].

Reduced graphene oxide (rGO) has been frequently employed as a carbon-based catalyst support owing to its large specific surface area and remarkable thermal and chemical stability [32,33]. The high specific surface area of rGO is beneficial for the loading of nanoparticles, thus improving the dispersion of particles on rGO nanosheets [34]. Moreover, many studies have demonstrated the presence of a synergistic effect between rGO and the anchoring particles, which can significantly enhance the catalytic performance [35,36]. For example, Co_3_O_4_ nanocrystals grown on rGO showed surprisingly high catalytic activity for oxygen reduction reaction and oxygen evolution reaction because of the synergetic chemical coupling effects between Co_3_O_4_ nanocrystals and graphene [37]. Owing to the cooperative effects of high surface area of graphene oxide sheets, graphene oxide-Ag_2_CO_3_ composites displayed a better photocatalytic activity for organic dyes degradation than pure Ag_2_CO_3_ crystal [38]. CoOOH-graphene nanosheets could act as an efficient oxygen evolution reaction electrocatalyst owing to the strong interface electron coupling between CoOOH nanosheets and graphene [39]. rGO-supported NiCo_2_O_4_ nanoparticles exhibited excellent catalytic property for the reduction of p-NP due to the synergistic effect between rGO and NiCo_2_O_4_ nanoparticles [40].

Herein, pure CoOOH nanorings and the composite of rGO and CoOOH nanorings were synthesized using a simple precipitation–oxidation–reduction method. Various characterization techniques such as XRD, TEM, SEM, BET and XPS were employed to investigate the composition, morphology and texture properties of the obtained nanocomposite. The catalytic performance of pure CoOOH nanorings and CoOOH/rGO nanocomposite was evaluated by the reduction of p-NP to p-AP employing excess NaBH_4_ as a reducing agent. Experimental results indicated that the catalytic activity of CoOOH/rGO nanocomposite was significantly enhanced compared with that of pure CoOOH nanorings due to the synergistic effect between the highly dispersed CoOOH nanorings and the unique surface properties of the rGO support. CoOOH nanorings and their composite were successfully synthesized and used as a catalyst for the reduction of p-NP, which has not been reported before. The results of this work demonstrate that the CoOOH/rGO nanocomposite is a potential catalyst for the reduction of p-NP to p-AP by NaBH_4_.

## 2. Materials and Methods

### 2.1. Materials

Cobalt chloride (CoCl_2_·6H_2_O), sodium hydroxide (NaOH), hydrogen peroxide (H_2_O_2_, 25 wt%), sodium borohydride (NaBH_4_) and p-nitrophenol (C_6_H_5_NO_3_) were purchased from Aladdin Industrial Corporation (Shanghai, China) with analytical grade. Graphene oxide (dispersed in water, 2 mg/mL, sheet thickness 0.7–1.2 nm) synthesized using the improved Hummer’s method was purchased from Suzhou Tanfeng Graphene Tech. Inc. (Jiangsu, China) The corresponding graphene oxide powder had a specific surface area of 1000–1217 m^2^/g. All chemicals were used as received without further purification. Deionized water was employed throughout.

### 2.2. Synthesis of CoOOH/rGO Nanocomposite

CoCl_2_·6H_2_O (1.25 mmol) and NaOH (25 mmol) were dissolved into 50 mL and 100 mL deionized water, respectively. The CoCl_2_ aqueous solution was added into 50 mL of 2 mg/mL graphene oxide suspension in water. The resulting mixture was sonicated for 2 h at room temperature. Then, the mixture was transferred to a three-necked flask containing 50 mL water at 50 °C under N_2_ atmosphere. After mechanical stirring for 3 h, the NaOH aqueous solution was added into the mixture. A pink precipitate was obtained and kept at 50 °C for 2 h. Then, N_2_ flow was stopped and 147 mmol H_2_O_2_ (25 wt%) was added into the mixture solution. A black brown suspension was obtained and kept at 50 °C for 2 h. After being cooled naturally to room temperature, the black brown precipitate was collected, washed with deionized water for several times and then dried at 80 °C for 6 h. Subsequently, the obtained powder was soaked in NaBH_4_ aqueous solution (0.1 mol/L) at 30 °C for 40 min; then, the suspension was centrifuged, washed with deionized water for several times and dried at 80 °C for 6 h. For the sake of comparison, pure CoOOH was also synthesized using the same preparation method without the addition of graphene oxide suspension.

### 2.3. Catalyst Characterization

The powder X-ray diffraction (XRD) patterns were carried out on a PANalytical X’pert-Pro powder X-ray diffractometer (PANalytical, Almelo, The Netherlands) with Cu Ka monochromatized radiation (λ = 0.1541 nm), using a scanning rate of 5 °/min. The specific surface areas, pore volumes and pore sizes of the products were tested with the commonly used BET method on a Quadrasorb SI instrument (Quantachrome, Boynton Beach, FL, USA) using nitrogen adsorption isotherms at 77 K. The morphologies of the products were collected on the JEOL JEM-2100 transmission electron microscopy (TEM, JEOL, Tokyo, Japan) with the accelerating voltage of 120 kV, as well as the JEOL JSM-7800F scanning electron microscopy (SEM, JEOL, Tokyo, Japan) with the accelerating voltage of 3 kV. X-ray photoelectron spectroscopy (XPS, Thermo Fisher Scientific, Waltham, MA, USA) analysis was performed on an ESCALAB 250Xi system with Al Ka as the X-ray source. The UV–Visible spectra were collected on a UV–Visible double-beam spectrophotometer (UV-4802, UNICO, Shanghai, China), with a wavelength range of 200 to 800 nm and a quartz cuvette with light path of 1 cm wide.

### 2.4. Catalytic Reduction of p-NP to p-AP by NaBH_4_

The reduction of p-NP to p-AP by NaBH_4_ was performed under controllable conditions to evaluate the catalytic performance of the obtained nanocomposite. Typically, the catalytic reaction was conducted at room temperature (ca. 23.5 °C) using a 3.5 mL standard quartz cell. p-NP aqueous solution (2.0 mL, 0.175 mmol/L) was first mixed with a freshly-prepared NaBH_4_ solution (0.7 mL, 0.05 mol/L) in the quartz cell. Then, the catalytic reaction was started by the quick addition of 0.1 mL suspension (2 g/L) containing the obtained CoOOH/rGO nanocomposite or pure CoOOH nanorings. The reduction process was monitored by UV–vis adsorption spectra of p-NP anions (400 nm) at regular intervals.

## 3. Results and Discussion

### 3.1. Characterization of Catalyst

Figure 1 shows the wide-angle XRD patterns of pure CoOOH and the composite of CoOOH with rGO. According to JCPDS 00-007-0169, every diffraction peak in the XRD pattern of pure CoOOH can be indexed to the characteristic diffraction peaks of the rhombohedral phase of CoOOH [41]. The diffraction peaks at 20.24°, 37.02°, 38.93°, 50.67°, 65.42° and 69.25° in the XRD pattern of pure CoOOH are readily indexed to the (0 0 3), (1 0 1), (0 1 2), (0 1 5), (1 1 0), and (1 1 3) facets of rhombohedral CoOOH, respectively. It is obvious that the characteristic diffraction peaks of the composite of CoOOH with rGO are identical to those of pure CoOOH except for the different intensities, and no conventional stacking peak of rGO was detected in the XRD pattern, implying that the diffraction intensity of rGO nanosheets is relatively low [42]. The different diffraction intensities suggest that the crystallinity of CoOOH decreased slightly during the formation of CoOOH/rGO composite.

The morphology and microstructure of the as-synthesized pure CoOOH and CoOOH/rGO nanocomposite were investigated using a scanning electron microscopy (SEM) and transmission electron microscopy (TEM) (Figure 2). As shown in Figure 2a,b, the obtained pure CoOOH has a regular hexagonal nanoring structure, and each nanoring has a large hollow interior. The particle size distribution of CoOOH nanorings is uniform, with an average outer diameter and edge length of ∼150 nm and 95 nm, respectively. A small number of CoOOH nanorings were broken, and the broken nanoring fragments can be observed in the images. SEM images of the CoOOH/rGO nanocomposite indicate that the as-synthesized composite is composed of aggregated and crumpled rGO nanosheets along with lots of CoOOH nanorings, many of which were wrapped together by rGO nanosheets (Figure 2c,d). It is obvious that CoOOH nanoring’s structure was well maintained during loading with rGO nanosheets. The TEM image shows that CoOOH nanorings were decorated on the rGO nanosheets successfully (Figure 2e). A high resolution TEM image of the nanocomposite shows an interplanar spacing between adjacent planes of 0.44 nm, which is consistent with the (0 0 3) crystal plane of rhombohedeal CoOOH (Figure 2f). These images demonstrate that rGO nanosheets can be anchored by CoOOH nanorings through the current facile precipitation–oxidation–reduction method.

The nitrogen adsorption–desorption isotherms and BJH adsorption pore size distributions of pure CoOOH and CoOOH/rGO nanocomposite are plotted in Figure 3. Type IV isotherms (IUPAC classification) are present for both pure CoOOH and CoOOH/rGO nanocomposite, indicating their mesoporous characteristics (Figure 3a) [43]. The pure CoOOH exhibits an H_1_-type hysteresis loop, which is related to the capillary condensation inside the mesopores, indicating the existence of mesoporous structure. An H_2_ hysteresis loop was observed for CoOOH/rGO nanocomposite, implying its complex pore structure. In addition, the hysteresis loop of CoOOH/rGO is significantly larger than that of pure CoOOH, suggesting the increased amount of pores in the nanocomposite [44]. It is thus likely that some of the pores in the nanocomposite were generated by the voids of aggregated particles, given that many of the CoOOH nanorings were wrapped together by rGO nanosheets. The pore size distribution derived from the adsorption branch of pure CoOOH shows a broad distribution, ranging from 1.7 nm to 61.9 nm, while that of CoOOH/rGO nanocomposite is broader, ranging from 1.9 nm to 93.6 nm (Figure 3b). The specific surface area, pore volume and pore diameter of pure CoOOH and CoOOH/rGO nanocomposite based on BET method are depicted in Table 1. The BET specific surface areas of pure CoOOH and CoOOH/rGO nanocomposite are 72.6 and 68.2 m^2^·g^−1^, respectively. The slight decrease in the nanocomposite specific surface area may be due to the stacking, coating and curling between CoOOH nanorings and rGO nanosheets. The pore volume of CoOOH/rGO nanocomposite is larger than that of pure CoOOH (being 0.34 and 0.29 cc·g^−1^, respectively), which is consistent with more pores in the nanocomposite. Moreover, the average pore diameters of pure CoOOH and CoOOH/rGO nanocomposite are basically the same (being 3.5 and 3.4 nm, respectively), implying that the pores in the nanocomposite were mainly from CoOOH nanorings. This also proves that the structure of CoOOH nanorings was well preserved in the nanocomposite.

The chemical composition and surface chemical states of pure CoOOH and CoOOH/rGO nanocomposite were qualitatively characterized with X-ray photoelectron spectroscopy (XPS). The results suggest that the main peaks in the XPS full spectra of pure CoOOH and CoOOH/rGO nanocomposite can be ascribed to Co, O or C elements without any impurities (Figure 4a). In general, the presence of C element in the spectrum of pure CoOOH can be attributed to the hydrocarbon contaminants inherent in XPS analysis. It is obvious that the peak intensity of C 1s spectrum of CoOOH/rGO nanocomposite is higher than that of pure CoOOH, while the peak intensities of Co and O 1s spectra are lower than those of pure CoOOH. These can be attributed to the presence of rGO nanosheets on the surface of CoOOH/rGO nanocomposite. The C 1s spectrum of CoOOH/rGO nanocomposite observed at 284.6 eV can be fitted into three component peaks at 284.5, 286.3, and 288.2 eV (Figure 4b). The major peak located at 284.5 eV corresponds to the *sp*^2^ graphitic carbon (C-C), indicating that most C atoms were still in the honeycomb lattice of rGO and the structure of rGO was preserved. While the other two peaks can be attributed to the C-O bond carbon (286.3 eV) and carbonyl carbon (C=O, 288.2 eV), resulting from the oxygen-contained carbon atoms, which is consistent with the low content of oxygen-containing functional groups in rGO. Detailed information on the chemical oxidation states can be obtained from the high resolution spectra of the detected elements. As shown in Figure 4c, the Co 2p spectrum of CoOOH/rGO nanocomposite presents two main peaks with binding energies of 780.2 and 795.3 eV, corresponding to Co 2p_3/2_ and Co 2p_1/2_, respectively. Meanwhile, the binding energies of the two peaks in the Co 2p spectrum of pure CoOOH are 780.1 and 795.1 eV, respectively. The shift of binding energies towards higher values (by 0.1–0.2 eV) when CoOOH/rGO nanocomposite was formed is mainly due to the interaction between the CoOOH nanorings and rGO support. The peak intensity of Co 2p in CoOOH/rGO is significantly weaker than that of pure CoOOH, which indirectly shows that the CoOOH nanorings were wrapped by rGO nanosheets. Moreover, the curve-fitted Co 2p_3/2_ spectrum of CoOOH/rGO has two peaks located at 780.0 and 781.5 eV, and the curve fitting peaks of pure CoOOH are at 780.1 and 781.8 eV, which are the characteristic peaks of CoOOH. Furthermore, the high-resolution O 1s spectrum of CoOOH/rGO nanocomposite observed at 531.3 eV can be deconvoluted into four typical peaks at 529.4, 530.8, 532.0 and 532.8 eV, corresponding to the metal oxygen bond (Co-O), hydroxyl species (i.e., CoOOH and C-OH), C-O bonds, and surface bonded water, respectively (Figure 4d). The peak area at 530.8 eV is evidently larger than the peak at 529.4 eV, indicating the presence of C-OH bonds in the nanocomposite, which in turn proves that the CoOOH/rGO is composed of CoOOH nanorings and rGO. The O 1s spectrum of pure CoOOH can be fitted into three peaks at 529.6, 530.7 and 531.8 eV, which can be assigned to the Co-O bond, Co-OH bond in CoOOH and surface adsorbed water, respectively. The XPS results together with TEM and SEM images unequivocally demonstrate the successful immobilization of CoOOH nanorings on the surface of rGO surface.

### 3.2. Catalytic Reduction of p-Nitrophenol

The catalytic reduction of p-NP to p-AP with excessive NaBH_4_ was used as a probe reaction to evaluate the catalytic performance of the obtained samples. The reaction was performed in a standard quartz cell and the time-dependent concentration change of the reactant (p-NP) and product (p-AP) was monitored by an in situ UV–vis spectrophotometer. The light yellow p-NP aqueous solution has a strong characteristic absorption peak at 317 nm. After adding NaBH_4_ aqueous solution, the color of the mixture solution turned to green-yellow and the adsorption peak shifted to 400 nm due to the formation of p-nitrophenolate ions under alkaline condition. According to the standard electrode potential (p-NP/p-AP = −0.76 V, H_3_BO_3_/BH_4_^−^ = −1.33 V), the reduction of p-NP to p-AP is considered to be a thermodynamically spontaneous process. However, the reaction cannot proceed without an efficient catalyst due to the high kinetic barrier and potential difference between the mutually repelling negative donor (BH_4_^−^ ions) and acceptor (p-nitrophenolate ions) [45]. With the addition of CoOOH/rGO nanocomposite, the solution changed from green-yellow to colorless quickly. The time-dependent absorption spectra show that the absorption peak of p-NP at 400 nm decreased gradually with time, and a new absorption peak located at 300 nm simultaneously appeared, indicating the reduction of p-NP to p-AP (Figure 5a). The adsorption peak at 400 nm no longer decreased after 3.25 min, suggesting the complete reduction of p-NP. When the pure CoOOH was used as catalyst, it took about 10.42 min for the full conversion of p-NP to p-AP (Figure 5b). It is obvious that the decreasing rate of the peak intensity at 400 nm varies greatly as the reaction proceeds, implying different catalytic activities of CoOOH/rGO and pure CoOOH in the reduction of p-NP (Figure 5a,b).

The reaction kinetics of the catalytic reduction of p-NP to p-AP using NaBH_4_ as a reducing agent has been frequently studied. In this work, the initial molar ratio of NaBH_4_ to p-NP was set at 100 (i.e., in the presence of a significant excess of NaBH_4_). Due to the high initial concentration of NaBH_4_, the concentration of BH_4_^−^ remained basically stable during the reaction, and thus hardly affects the reaction rate. Then, it is reasonable to suppose that the reduction reaction follows a typical pseudo-first-order kinetics. The apparent rate constant (*k*_app_) can be calculated from the following equation:*−ln*(*C*_t_
/*C*
_0_) = −*ln*(*A*
_t_
*/A*
_0_) = *k*_app_*t*
(1)

where *A*_t_ and *A*_0_ are the absorbance of p-NP (at 400 nm wavelength) at time of *t* and 0 min, respectively. *C*_t_ and *C*_0_ represent the corresponding concentrations of p-NP acquired from *A*_t_ and *A*_0_, respectively. Therefore, the apparent rate constant values are determined by the slopes of the respective linear curves of *t* versus −*ln*(*A*_t_/*A*_0_). Compared with pure CoOOH, CoOOH/rGO nanocomposite could catalyze the reduction of p-NP to p-AP in a much shorter time (Figure 6a). The apparent rate constant values for the reduction reaction when using 0.2 mg of CoOOH/rGO nanocomposite or pure CoOOH are 1.77 and 1.22 min^−1^, respectively (Figure 6b). Furthermore, we conducted three replicate experiments with CoOOH/rGO as the catalyst at the same condition, and the experimental error was about 6.7%, which can be attributed to the difference in ambient temperature and other experimental uncertainties. This also shows good experimental reproducibility. Figure 6 demonstrates that the CoOOH/rGO nanocomposite has a higher catalytic activity, despite its specific surface area being slightly lower than that of pure CoOOH (Table 1). It has been proven that the reduction of p-NP to p-AP is a surface-controlled process, which can be explained well by the Langmuir–Hinshelwood mechanism [7]. This suggests that the reduction process takes place on the catalyst surface and the surface area would have a great effect on the apparent rate constant, accordingly. Since the slight reduction in surface area observed in CoOOH/rGO nanocomposite did not render a decrease in its catalytic activity, the much improved activity of CoOOH/rGO nanocomposite must be due to some other features related to its structure [46].

For heterogeneous catalytic reaction, the catalytic reduction of p-NP occurs only when BH_4_^−^ and p-NP are adsorbed on the surface of the catalyst. The CoOOH/rGO nanocomposite served as the catalyst to relay active hydrogen species and electrons from the adsorbed donor (BH_4_^−^) to the adsorbed acceptor (p-NP) nearby, thereby triggering the reduction reaction. Therefore, the efficient adsorption performance for p-NP and BH_4_^−^ as well as its excellent electron transport ability, are the key factors for a catalyst to have superior activity [36]. In addition, Qu et al. have demonstrated that, for the catalytic hydrogenation of p-NP, increasing the amount of reactants adsorbed on the catalyst surface is beneficial in improving the catalytic performance [47]. Therefore, the superior catalytic performance of CoOOH/rGO nanocomposite can be ascribed to the synergistic effect between the highly dispersed CoOOH nanorings and the unique surface properties of the rGO support, which can be explained as follows. p-NP ions were preferentially adsorbed on the surface of rGO nanosheets via π–π stacking interaction, leading to a high concentration of p-NP near the CoOOH nanorings anchored on rGO nanosheets, which was conducive to the efficient contact between p-NP and the CoOOH nanorings, thereby improving the catalytic activity. Moreover, the electron transfer from rGO nanosheets to CoOOH nanorings increased the local electron density at their interface, facilitating the uptake of electrons by p-NP [2,48,49,50,51].

Moreover, it is clear that there is an induction time (*t*_0_) in the catalytic reduction of p-NP, during which no reduction occurs. The existence of an induction time has been widely reported. Some assumptions about the induction time, including spontaneous surface reconstruction, activation time and diffusion time, have been discussed. However, the concrete reasoning remains unclear and a controversial issue [6]. Generally, the higher the catalytic activity of a catalyst, the shorter the induction time [20]. When used as a catalyst, CoOOH/rGO nanocomposite has an induction time of less than 0.95 min, while pure CoOOH exhibits an induction time of about 6 min (Figure 6a,b, Table 2). The CoOOH/rGO nanocomposite has high catalytic activity and correspondingly short induction time, which is consistent with the general findings. The difference in induction time may be attributed to the different diffusion time of reactants to the catalyst surface. For a heterogeneous catalytic reaction, the reactants first diffuse from the bulk solution to the catalyst surface, and then adsorb onto the catalyst surface before the reaction occurs. From the above discussion, CoOOH/rGO nanocomposite had high adsorption ability towards p-NP because of the strong π−π interaction. Moreover, the reduction process by NaBH_4_ aqueous solution can remove part of the oxygen-containing functional groups on the surface of graphene oxide, further enhancing the π–π interaction between p-NP and rGO nanosheets, which enabled a high concentration of p-NP near the CoOOH nanorings on rGO surface. The strong π−π interaction between p-NP and rGO nanosheets promoted the faster diffusion of p-NP molecules to the CoOOH/rGO surface, which ultimately led to a shorter induction time.

A comparative study for the reduction of p-NP to p-AP with other catalysts is given in Table 3. For the Langmuir–Hinshelwood model reaction, the amount of catalyst would have a large effect on the apparent rate constant (*k*_app_) by affecting the total surface area. To eliminate the influence of catalyst dosage on the catalytic activity, the mass-normalized rate constant (*k*_nor_) is usually adopted to compare the activities of various catalysts [52], as defined as Equation (2):*k*_nor_ = *k*_app_/*w*_cat_(2)
where *w*_cat_ is the mass concentration of the catalyst in the solution. In addition, Table 3 shows the concentrations of p-NP and NaBH_4_ (i.e., *C*_p-NP_ and *C*_NaBH4_, respectively) used in the catalytic reaction process, which is also an important factor that affects the apparent rate constant. It can be seen from Table 3 that the catalytic activity of the as-prepared CoOOH/rGO nanocomposite in this work is comparable or even better than that of supported noble metal catalysts (e.g., graphene/Au, Pt@Ag, rGO-Ag and Pd/graphene), and is obviously higher than that of cobalt based composites (e.g., NiCo_2_O_4_/rGO, CoO_x_/CN and Co_3_O_4_@C), confirming its relatively high efficiency in the catalytic reduction of p-NP.

Activation energy (*E_a_*) is an important kinetic parameter to evaluate the performance of a catalyst, which reflects the temperature dependence of the (apparent) rate constant during the catalytic process. The catalytic reduction of p-NP over the CoOOH/rGO nanocomposite at different reaction temperatures was investigated and the corresponding *k*_app_ values were studied. The results are shown in Figure 7. It can be clearly seen that the increase in reaction temperature results in a larger *k*_app_ (Figure 7a), indicating the enhanced catalytic activity of CoOOH/rGO nanocomposite, which can be ascribed to the increase in molecular motion rate of the reactants as well as the increase in collision probability [62,63]. When the temperature was increased from 13.5 °C to 23.5 °C, the value of *k*_app_ increased from 1.03 min^−1^ to 1.77 min^−1^. The apparent activation energy of the catalytic reduction process can be obtained according to the Arrhenius equation:ln*k*_app_ = ln*A* − *E_a_*/*RT*(3)
where *A* is the pre-exponential factor and *R* is the universal gas constant (8.314 J·mol^−1^·K^−1^), and *T* is the temperature in K. Figure 7b shows that ln*k* has a good linear relationship with 1/*T* (R^2^ = 0.9995) and the apparent activation energy for the reduction of p-NP was calculated to be 38.18 kJ·mol^−1^ over the CoOOH/rGO nanocomposite. This value is comparable to the previously reported values for the supported noble metal catalysts [64,65].

Besides the catalytic reactivity, the recyclability is also a vital aspect to evaluate the performance of a catalyst. The reusability of the as-prepared nanocomposite was assessed by continuously adding concentrated p-NP aqueous solution after each cycle to adjust the concentration of p-NP to 0.125 mmol/L before reaction. The concentration of NaBH_4_ in the first cycle was 12.5 mmol/L, which was 100 times that of p-NP, and no NaBH_4_ was added in subsequent cycles. It can be observed from Figure 8 that the CoOOH/rGO nanocomposite exhibits satisfactory stability during five consecutive cycles. As the number of cycles increased, although the catalytic activity of the CoOOH/rGO nanocomposite decreased slightly, p-NP could still be completely reduced to p-AP within 6 min in the fifth cycle. Moreover, the structural characteristics of the CoOOH/rGO nanocomposite after the catalytic reduction of p-NP to p-AP were studied (Figure 8b). CoOOH nanorings and rGO nanosheets can also be clearly observed, indicating that the structure of CoOOH/rGO nanocomposite was maintained to a certain or large extent during the reduction reaction. However, compared to the CoOOH/rGO nanocomposite before the reduction reaction, more nanorings were broken, and broken nanoring fragments could be observed. This possibly contributed to the catalyst deactivation. In addition to the slow deactivation of the catalyst itself, which led to the reduction in catalytic activity, the consumption and decomposition of NaBH_4_ are also part of the reason for the reduced catalytic activity. The results indicate that the CoOOH/rGO nanocomposite possesses superior activity and good stability for the catalytic reduction of p-NP to p-AP with excessive NaBH_4_.

## 4. Conclusions

The CoOOH/rGO nanocomposite has been successfully fabricated using a facile precipitation–oxidation–reduction one-step method. The CoOOH nanorings with a hexagonal outer profile were uniformly anchored on the surface of rGO nanosheets and the obtained CoOOH/rGO nanocomposite exhibited superior activity in the catalytic reduction of p-NP to p-AP with excessive NaBH_4_. It only took 3.25 min for the complete conversion of p-NP over the CoOOH/rGO nanocomposite at room temperature, and the apparent rate constant and apparent activation energy were estimated to be 1.77 min^−1^ and 38.18 kJ·mol^−1^, respectively. Additionally, the nanocomposite showed good stability in five successive cycles. The catalytic activity of CoOOH was significantly increased by immobilizing CoOOH nanorings on the surface of rGO, due to the synergistic effect between CoOOH nanorings and rGO nanosheets. Considering its easy preparation, excellent activity and good recyclability, the nanocomposite of CoOOH nanorings with rGO would be a promising catalyst for the catalytic reduction of p-NP to p-AP with NaBH_4_.

## Figures and Tables

**Figure 1 materials-15-08862-f001:**
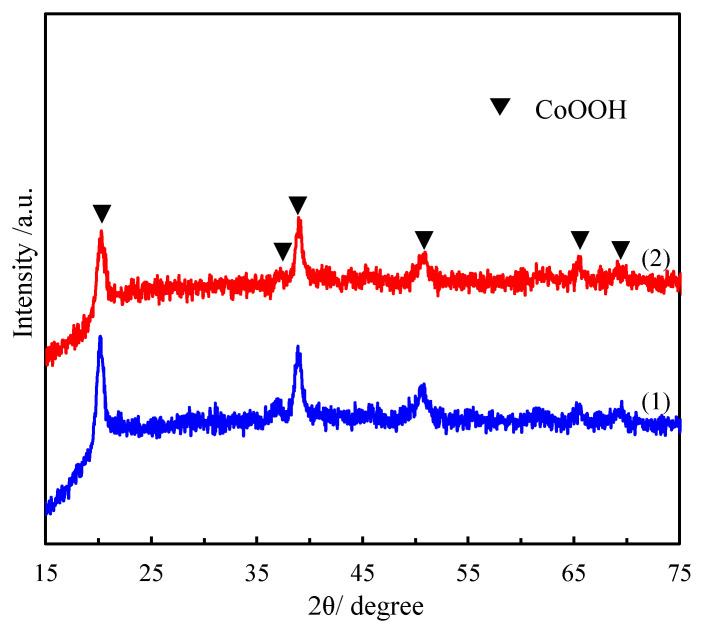
XRD patterns of (1) pure CoOOH and (2) CoOOH/rGO.

**Figure 2 materials-15-08862-f002:**
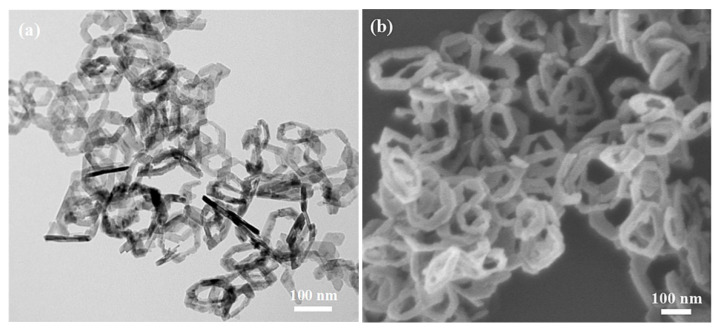
(**a**) TEM and (**b**) SEM images of pure CoOOH. (**c**,**d**) SEM images of CoOOH/rGO at different magnifications. (**e**,**f**) TEM images of CoOOH/rGO at (**e**) low magnification and (**f**) high magnification.

**Figure 3 materials-15-08862-f003:**
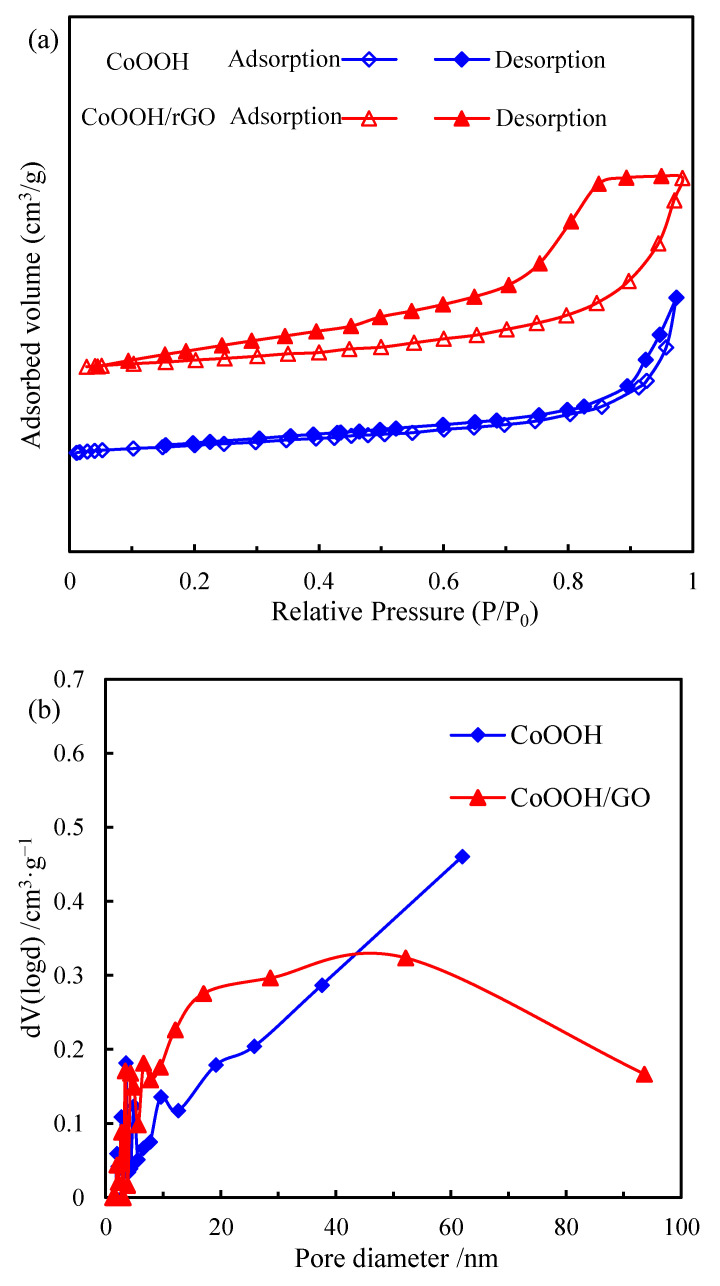
(**a**) N_2_ adsorption-desorption isotherm curves and (**b**) BJH pore size distributions of pure CoOOH and CoOOH/rGO nanocomposite.

**Figure 4 materials-15-08862-f004:**
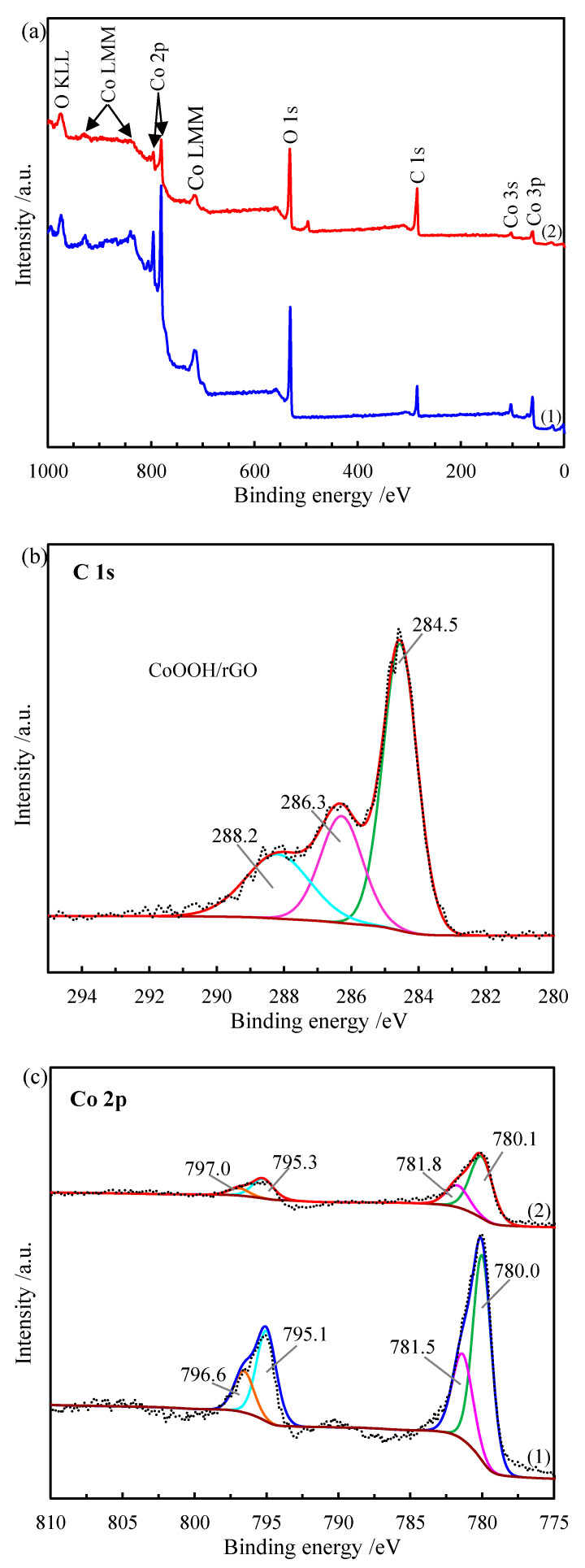
(**a**) XPS survey spectra of (1) pure CoOOH and (2) CoOOH/rGO. (**b**) C 1s spectra of CoOOH/rGO. (**c**) Co 2p spectra and (**d**) O 1s spectra of (1) pure CoOOH and (2) CoOOH/rGO.

**Figure 5 materials-15-08862-f005:**
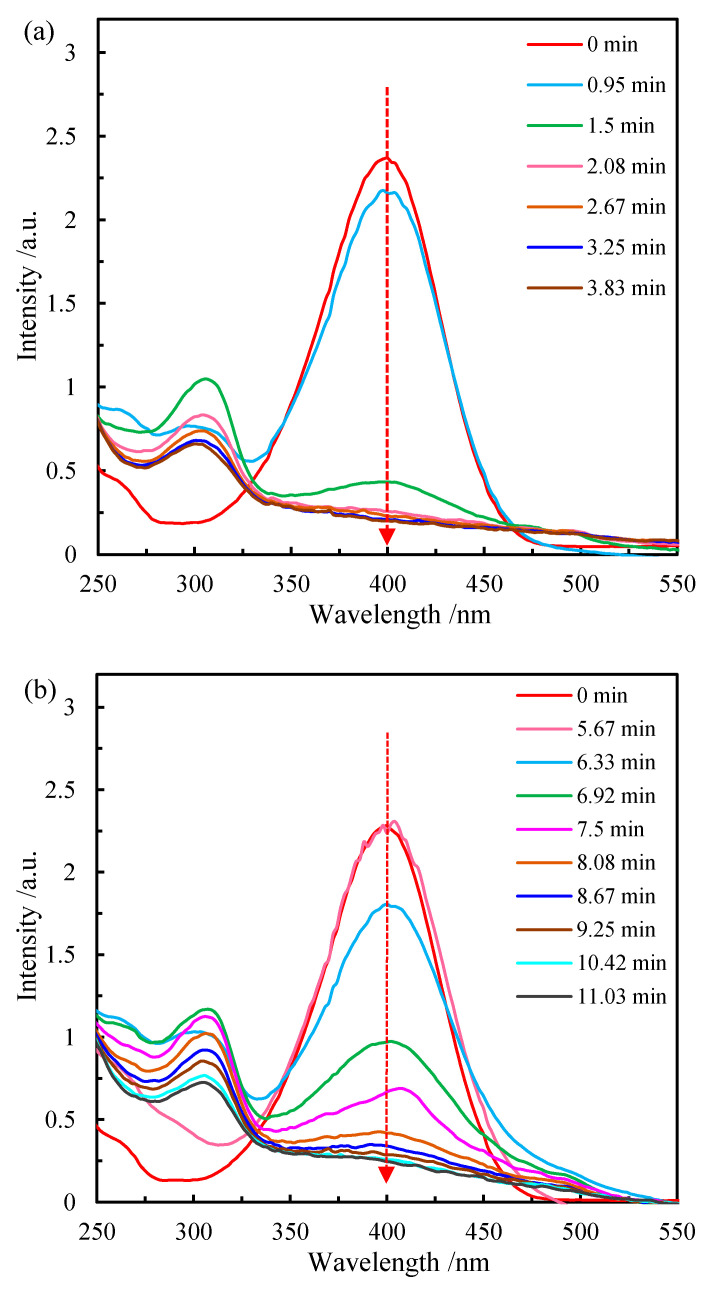
The UV–vis absorption spectra during the catalytic reduction of p-NP with (**a**) CoOOH/rGO and (**b**) pure CoOOH at different reaction times. Reaction conditions: Initial p-NP concentration (*C*_p-NP_) at 0.125 mmol/L, molar ratio of NaBH_4_ to p-NP at 100, catalyst weight (*m*_cat_) at 0.2 mg.

**Figure 6 materials-15-08862-f006:**
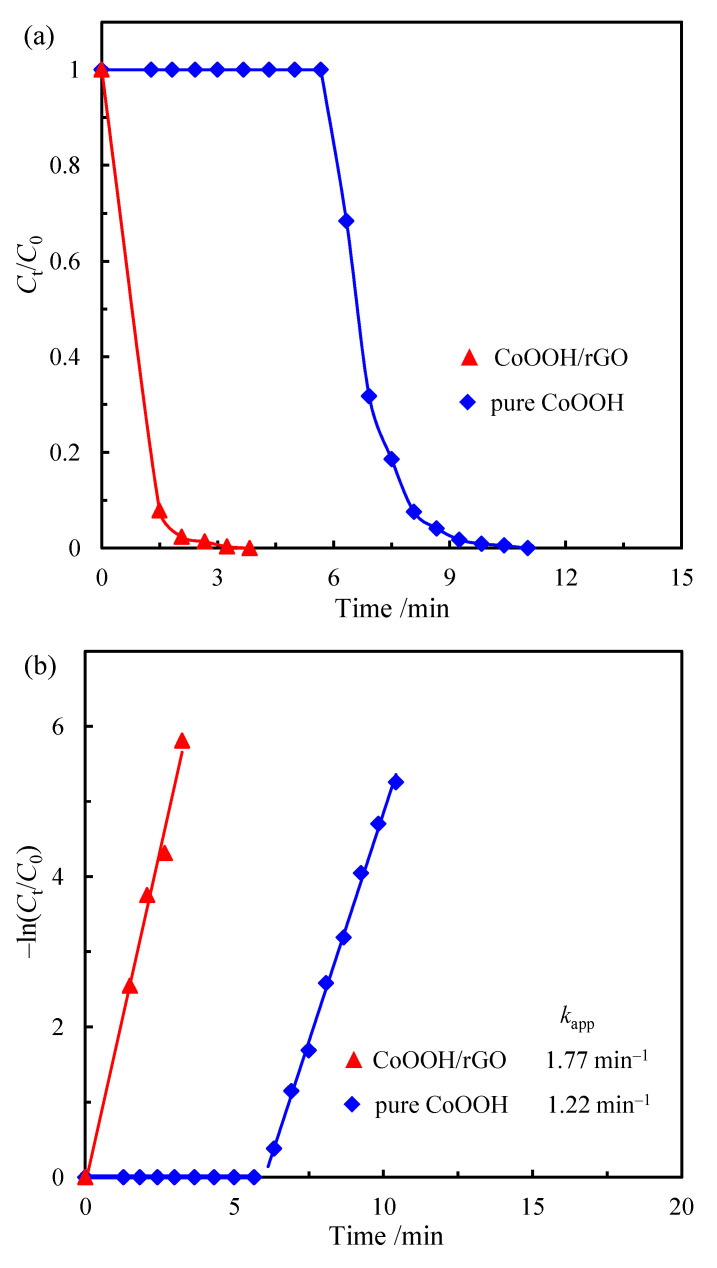
(**a**) *C*_t_/*C*_0_ as a function of the reaction time and (**b**) kinetic analysis over pure CoOOH and CoOOH/rGO.

**Figure 7 materials-15-08862-f007:**
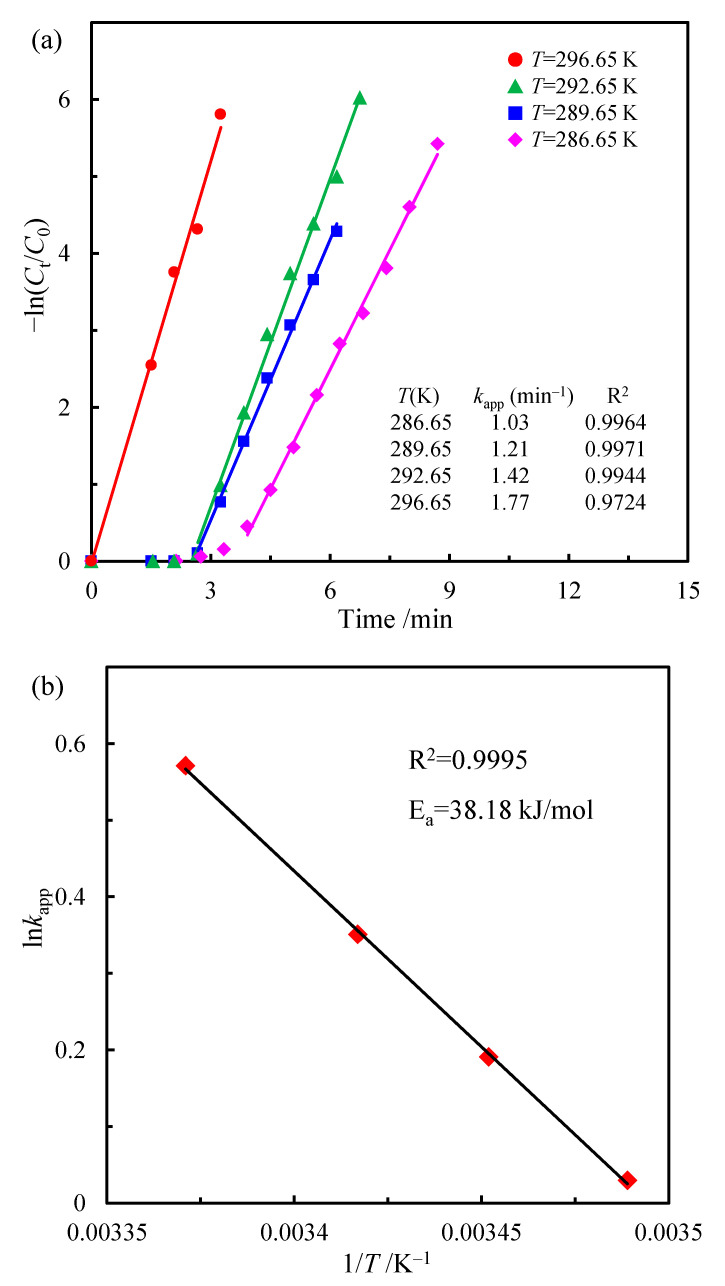
(**a**) Plots of −ln (C_t_/C_0_) versus the reduction time for the catalytic p-NP reduction at different temperatures over CoOOH/rGO. (**b**) The corresponding Arrhenius plot at different temperatures.

**Figure 8 materials-15-08862-f008:**
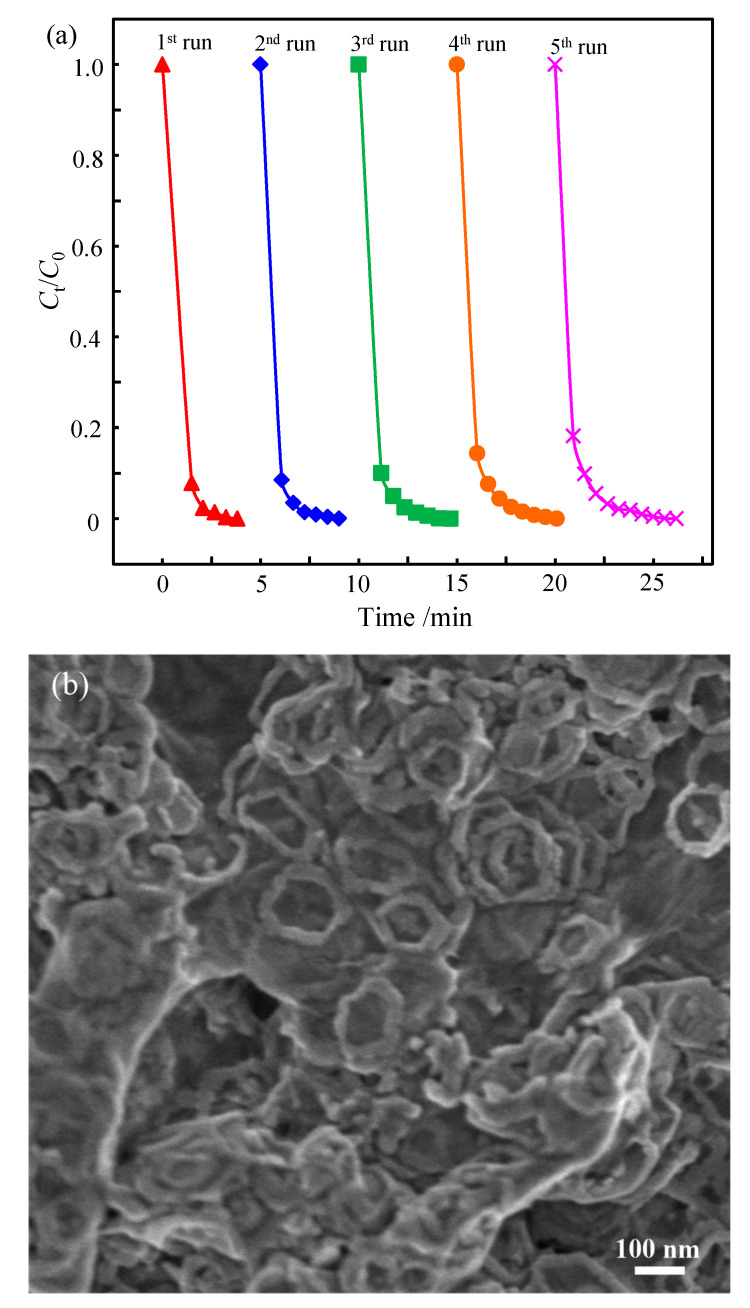
(**a**) The reusability of CoOOH/rGO for the catalytic reduction of p-NP. (**b**) SEM image of CoOOH/rGO after the reduction of p-NP.

**Table 1 materials-15-08862-t001:** The surface properties of pure CoOOH and CoOOH/rGO.

Samples	BET Surface Area (m^2^/g)	Total Pore Volume (cc/g)	Average Pore Diameter (nm)
pure CoOOH	72.6	0.29	3.5
CoOOH/rGO	68.2	0.34	3.4

**Table 2 materials-15-08862-t002:** The induction time of CoOOH/rGO and pure CoOOH in the reduction of p-NP ^a^.

Catalyst	*t*_0_/min
CoOOH/rGO	*t*_0_ < 0.95
pure CoOOH	5.67 < *t*_0_ < 6.33

^a^ Based on the reaction results in Figure 6.

**Table 3 materials-15-08862-t003:** The mass-normalized rate constants of various catalysts.

Catalyst	*C* _p-NP_	*C* _NaBH4_	*w* _cat_	*k* _app_	*k*_nor_ ^a^	Reference
	mM	mM	g/L	min^−1^	L·min^−1^·g^−1^	
Au (small)	0.19	1.37	0.003	0.15	44.82	[8]
Au (large)	0.19	1.37	0.003	0.03	7.28	[8]
Pt@Ag	0.09	0.01	0.016	0.36	21.66	[53]
Pd	0.07	1.65	0.046	0.73	15.80	[10]
Graphene/Au	1.41	139.53	0.465	2.00	4.30	[54]
Au/Co@N-C	0.23	23.43	0.031	2.25	71.92	[55]
Au/BNO	1.61	96.77	0.323	2.25	6.98	[56]
AuNi/MoS_2_	0.10	4.76	0.318	1.06	3.34	[3]
rGO-Ag	0.20	52.87	0.080	1.18	14.75	[42]
Ag/Co_3_O_4_	0.54	53.57	0.054	2.88	53.76	[57]
Ag-Co/CNFs	0.09	18.18	0.303	1.03	3.41	[58]
Pd/graphene	0.28	6.98	0.093	0.67	7.16	[35]
CoFe/rGO	0.09	10.00	0.667	4.61	6.92	[59]
Co/PCNS	0.09	40.98	0.033	0.31	9.60	[45]
Co/CoO/Co_3_O_4_-NC	0.14	0.67	0.202	0.66	3.29	[8]
NiCo_2_O_4_/rGO	0.58	16.67	0.083	0.71	8.56	[40]
CoO_x_/CN	0.32	40.00	0.080	0.25	3.15	[60]
Co_3_O_4_/HNTs	0.12	46.40	0.033	0.26	7.96	[30]
Co_3_O_4_@C	8.33	8.33	1.667	0.76	0.45	[31]
Co_3_O_4_	0.04	15.86	0.100	0.14	1.41	[61]
CoOOH/rGO	0.13	12.50	0.071	1.77	24.78	This work
CoOOH	0.13	12.50	0.071	1.22	17.08	This work

^a^ The mass-normalized rate constant calculated based on the catalyst amount including the support, as described in Equation (2).

## Data Availability

All data are available in the manuscript.

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
