# Peer review of "Facile Synthesis of CoOOH Nanorings over Reduced Graphene Oxide and Their Application in the Reduction of p-Nitrophenol"

_materials, 2022, doi:10.3390/ma15248862_

Round 1

Reviewer 1 Report

This manuscript deals with Photocatalytic activity of CoOOH/GO nanocomposite for the reduction of p-nitrophenol. However, some points in this manuscript should enhance tremendously. Therefore, I recommend a major revision.

1)                  The novelty of this study should be mentioned clearly in the introduction.

2)                  The optical properties of sample should by analyzed and discussed.

3)                  The standard - Degussa/Evonik P25 - should be used as a reference sample.

4)                  TOC - total organic carbon - should be measured at the end of the reaction.

5)                  Band potentials are very important for photocatalytic reactions. The authors should measure VB and CB potentials using Mott- Schottky analysis.

6)                  How about the control experiment such as photolysis in reduction of p-nitrophenol?

7)                  Error bar/experimental error should be provided for all the photocatalytic results.

8)                  The characterization of used catalyst should be provided in this manuscript.

Reviewer 2 Report

Article "Facile synthesis of CoOOH nanorings over graphene oxide and their application in the reduction of p-nitrophenol" by Huihui Chen et al. is devoted to an interesting studying a model catalytic reaction of reduction of 4-nitrophenol using a cobalt-based composite as a catalyst. The manuscript competently describes all the studies. However, below are the comments that the author should definitely take into account.

Experimental

For what purpose was the composite treated with sodium borohydride? This also leads to the reduction of graphene oxide. In addition, it becomes unclear whether all the studies cited refer to the CoOOH/GO composite BEFORE NaBH4 treatment? The diffraction pattern will be different after even partial reduction. If the key point is application in the reduction of p-nitrophenol, then the name should be changed to CoOOH nanorings over reduced graphene oxide and all research results should be supplemented.

Results and discussion

If using a composite or pure CoOOH without pre-treatment, is catalysis efficiency achieved? Probably yes. Only the reaction time will be significantly shorter, since NaBH4 will also be spent on the reduction of the composite. In addition, the authors still need to realize that catalytic reactions occur in the presence of graphene oxide or its reduced form in the composite. Rather, the reduction of graphene oxide with the removal of part of the oxygen-containing groups just leads to a higher strong π−π interaction between p-NP and GO nanosheets promoted the faster diffusion of p-NP molecules for a shorter induction time. The authors should pay attention to the fact that in Table 3, metal nanoparticles on reduced graphene oxide or inert carbon materials are used as effective catalysts! Graphene oxide is just a very reactive substance due to the large number of oxygen functional groups.

Regarding cyclic catalytic reactions, it should also be said that an excess of NaBH4 leads to the reduction of graphene oxide, so in this case, the manuscript should still talk about the successful use of CoOOH/rGO as a catalyst for the reduction of p-NP.

There are also minor comments on the text.

Links should be placed in square brackets.

To be completed with references to the use of metal particle catalysts for the conversion of 4-nitrophenol (row 56-57)

Row 136 “powder”

In paragraph 2.3, it is necessarily to add information on UV spectroscopy.

On Fig. 2 e, f on the signature is marked "graphene". What do authors mean - reduced graphene oxide or graphene oxide?

Round 2

Reviewer 1 Report

The authors have made sufficient modifications, and I suggest that this paper be accepted without further modification.

Reviewer 2 Report

The response of the authors is satisfactory. The article is recommended for publication in its present form.